# Electromotive Enhanced Drug Administration in Oncology: Principles, Evidence, Current and Emerging Applications

**DOI:** 10.3390/cancers14204980

**Published:** 2022-10-11

**Authors:** Jolene Wong Si Min, Nidda Saeed, Annelies Coene, Mieke Adriaens, Wim Ceelen

**Affiliations:** 1Department of GI Surgery, Ghent University Hospital, 9000 Ghent, Belgium; 2Department of Electromechanical, Systems and Metal Engineering, Faculty of Engineering and Architecture, Ghent University, 9000 Ghent, Belgium; 3Cancer Research Institute Ghent (CRIG), 9000 Ghent, Belgium; 4Department of Chemistry, Faculty of Sciences, Ghent University, 9000 Ghent, Belgium

**Keywords:** electromotive, electric-driven, iontophoresis, electroporation, drug transport

## Abstract

**Simple Summary:**

Since the 17th century, the use of electric currents to improve the transport of drugs into human tissues has been described. Currently, electrically driven drug transport is used in a variety of medicinal fields such as the urinary system, skin, eye and others. In this review, we summarize the principles and factors that govern electrically driven drug transport and discuss its current and emerging applications for the treatment of cancer patients.

**Abstract:**

Local-regional administration of cytotoxic drugs is an important adjunct to systemic chemotherapy amongst cancer patients. It allows for targeted delivery of agents at high concentration to target sites while minimizing systemic side effects. Despite the pharmacokinetic advantages of the local–regional approach, drug transport into tumor nodules remains limited due to the biophysical properties of these tissues. Electromotive enhanced drug administration (EMDA) represents a potential solution to overcome challenges in local drug transport by applying electric currents. Through electrokinetic phenomena of electromigration, electroosmosis and electroporation, electric currents have been shown to improve drug penetration and distribution in a wide variety of clinical applications. Amongst patients with non-muscular invasive bladder cancer (NMIBC) and basal and squamous cell skin cancers, EMDA has been successfully adopted and proven efficacious in several pre-clinical and clinical studies. Its application in ophthalmological and other conditions has also been explored. This review provides an overview of the underlying principles and factors that govern EMDA and discusses its application in cancer patients. We also discuss novel EMDA approaches in pre-clinical studies and explore future opportunities of developments in this field.

## 1. Introduction

The ideal drug delivery model is safe, convenient, site-specific and can maximize therapeutic efficacy while ensuring minimal toxicity to unintended sites [1]. As such, a local–regional approach is logical and has pharmacokinetic (PK) advantages when compared with systemic drug administration [2]. This approach has been most widely adopted in bladder and peritoneal cancers, where intra-vesical and intra-peritoneal (IP) delivery of cytotoxic chemotherapeutic agents have been shown to have superior anti-tumoral effects [3,4,5]. However, both delivery modes rely heavily on passive diffusion for the transport of drugs into target tissues—a process hindered by the relative impermeability of both urothelial and peritoneal membranes [6,7,8]. Furthermore, the penetration of drugs into tumor nodules with elevated interstitial fluid pressures (IFP) represents an additional barrier to efficient drug transport [9].

Recognizing the challenges in local–regional drug delivery, the use of electric forces to enhance drug penetration has gained increasing popularity in the recent decade [10,11]. First described by Veratti in 1747, the concept of electricity-enhanced drug transport is not new [12]. In the early 20th century, Leduc, through a classical experiment using two rabbits connected in an electrical series circuit with strychnine sulfate and potassium cyanide solutions, proved that ionized drugs could penetrate the skin and exert a systemic effect. He also demonstrated the importance of polarity with respect to an ionized drug and its counter ion [13]. This phenomenon was termed iontophoresis, describing an accelerated delivery of charged ions into tissues when an electric current was applied through a drug solution [14,15,16].

By the 1930s, iontophoresis was frequently adopted in ophthalmologic and skin conditions and was found to be highly effective in the treatment of hyperhidrosis [17,18]. It was also used to enhance the effects of local anesthesia to the tympanic membrane, oral mucosa, skin and the eye [19]. The term electromotive drug administration (EMDA) was subsequently coined in 1994, recognizing electroporation as a secondary force at play with iontophoresis that resulted in an increased penetration of both ionized and unionized drugs into surrounding tissues [20,21,22]. Since then, EMDA has been frequently used to describe electricity-enhanced intra-vesical administration of cytotoxic and anesthetic agents for urological conditions.

In this review, we summarize the underlying principles and factors influencing EMDA-enhanced drug delivery. In addition, we examine pre-clinical and clinical data in its application in a diverse group of oncological conditions, ranging from urological, skin, ophthalmology and others, emerging technologies and explore future opportunities for development in this field.

## 2. Fundamental Principles in EMDA

EMDA refers to the use of a low intensity electrical current to drive drugs across various tissues. There are three main electrokinetic phenomena that govern EMDA: electromigration, electroosmosis and electroporation (Figure 1) [21,22,23]. The summation of electro-migratory and electro-osmotic forces is termed iontophoresis and describes the mechanism in which the delivery of ionized drugs across biological membranes is enhanced through the application of a mild electric current in an electrolyte drug solution [23]. In electromigration, the repulsion of cations by the anode and anions by the cathode result in ionic fluxes across tissues to maintain electrical neutrality. This is coupled with electroosmosis, where solvent flows in an anode to cathode direction, providing a secondary driving force in the transport of cationic drugs [24]. This convective solvent flow also represents the main mechanism by which unionized drug compounds are transported across membranes [21]. The two main electrokinetic phenomena (electromigration and electroosmosis) in EMDA-mediated drug transport can be described by the modified Nernst–Planck equation (Appendix A) [25,26]. Finally, electroporation describes the formation of aqueous pores by the application of an electric current, thereby increasing the permeability of membranes and facilitating drug delivery [22,27]. This should be differentiated from “high voltage electroporation” techniques where voltages of >100 volts over very short durations (microseconds to milliseconds) are used to permeabilize the skin [28].

### 2.1. EMDA Devices

In transdermal applications, several Food and Drug Administration (FDA) approved EMDA devices exist; most use a current source and two electrode compartments, the latter consisting of an electrode immersed in an electrolyte (ionic conductor) solution or gel [29]. Both electrode compartments are placed at two distant sites on the skin while aiming for transdermal drug delivery [20]. The total set-up operates as an electrochemical cell (Figure 2):

In intra-vesical EMDA, a catheter-electrode is connected to a current generator, the *Physionizer^®®^ Mini 30N2* (Medolla, MO, Italy) and a positive polarity applied (Figure 3). The bladder is filled with a drug containing electrolyte solution such that if a cationic drug ion is present, penetration across the bladder wall is enhanced predominantly through an electromigratory phenomenon [24]. This contrasts with unionized large drug molecules, whose transport is more electro-osmotically driven [21].

### 2.2. Relationship between Current Intensity, Ion Valency and EMDA

Faraday’s law of electrochemical reaction states that [30]:$$Q=\frac{t_{i}It}{\left|Z_{i}\right|F}$$
where *Q* is the mass of drug delivered by electromotive transport, *t_i_* the transference number of the ionic species *i*, *I* the current applied (in amperes) and *t* the time duration (in seconds).

As such, the amount of drug delivered during EMDA is directly proportional to the current intensity and treatment time but inversely proportional to the charge of the drug ion. The transference number of a specific drug ion refers to its ability to carry electric current and is defined as the ratio of the electric current carried by the drug ion i to the total current carried by all ionic species within the electrolyte solution.

The linear relationship between applied current and electromotive transport was illustrated by Harding (1987), who found increasing rates of angiotensin release from Ringer’s solution with increasing amplitudes of electric current applied [31]. Amongst drug compounds with similar molecular weight, monovalent sodium ions were found to have a drug delivery efficiency that was more than twice that of divalent magnesium ions, reflecting the slower migration of drug ions with high valency when a constant electric current is applied [16]. This is also reflected by the inverse relationship between valency and drug delivery in Faraday’s equation above.

However, while increasing current can increase drug delivery in experimental models, thermal damage to healthy tissues is a significant concern in clinical applications. For example, in transdermal EMDA treatments, a current density exceeding 0.5 mA/cm^2^ induces skin irritation while a maximum of 15 to 20 mA is used in intravesical EMDA to prevent discomfort and tissue damage [10]. Other adjuncts adopted to reduce the likelihood of skin injury during transdermal EMDA include the use of well-saturated absorbent pads and ensuring that there is no direct contact between metallic components and the skin [32]. In addition, based on early transdermal experiments, pulsed direct current (DC) is preferred over continuous DC, as the latter has been found to cause skin polarization, which in turn results in skin irritation and likely a reduced drug delivery efficiency [33].

### 2.3. Relationship between Drug Physicochemical Properties and EMDA

The physicochemical properties of drug molecules and their carrier solution are important factors influencing EMDA [15]. The charge and molecular size of drug molecules and the pH and presence of buffer ions within its carrier solution affect iontophoresis. As only ionized or charged drugs may be delivered via electromotive forces, unionized drugs rely solely on electroosmosis for transport. Since the degree of drug ionization is often pH dependent, altering the pH of carrier solutions can have a significant impact during EMDA. In an experiment using lidocaine (a local anesthetic agent), Gangarosa et al. demonstrated that when an alkaline solution was added, conductivity was reduced by 15%, as an increase in pH drove the conversion of positively charged lidocaine ions to unionized molecules [34]. The impact of pH on EMDA delivery was further evaluated by Murthy et al., who found that iontophoretic flux of salicylic acid (SA) was significantly increased at higher pH due to the increased ionization of SA (at a pH of 1.2, SA is present solely in the unionized form and the only driving force for transport is electroosmosis. At a pH of 7.1, SA is completely ionized and electromigration drives drug transport) [35].

When present in carrier solutions, buffer ions act as competitors to charged drug ions, resulting in reduced iontophoretic delivery of the latter. As such, if EMDA is desired, the inclusion of small mobile ions in drug diluent solutions should be avoided.

### 2.4. Membrane or Barrier Properties and EMDA

The skin, urothelium and sclera are examples of biological barriers that can affect EMDA. In general, the porosity of membranes is affected by their thickness as well as pore size and charge. In transdermal EMDA, this membrane barrier is composed of up to 15 layers of corneocytes embedded within the intracellular lamellar lipid membrane, which make up the stratum corneum [36]. At physiological pH, the skin carries a net negative charge; hence, EMDA enhances the transport of positively charged drug ions with an anode while retarding the movement of negatively charged drug ions with a cathode. Neutral drug molecules whose transport is mainly driven by electroosmosis will experience enhanced transport with an anode due to the negatively charged skin barrier [32]. With transscleral applications, since the sclera also carries a net negative charge under physiological conditions, a similar pattern of iontophoretic flux is seen [37]. As the sclera is relatively porous when compared with the skin due to larger pore sizes, it allows for the easy penetration of macromolecules. An example is the fact that the transport of bevacizumab (molecular weight 149 kDa), an antibody targeting the vascular endothelial growth factor (VEGF), is enhanced 32-fold when 2 mA of current is applied during EMDA [38].

In intravesical oncological applications, the ideal penetration depth is to the lamina propria, which is at 1193 ± 26.9 µm [39]. When EMDA with mitomycin-C (MMC) is applied in an in vitro human bladder model, drug penetration at a constant current of 20 mA was highest at a depth between 80 and 200 µm and lowest at a depth between 2000 and 4000 µm [39,40]. This illustrates the importance of barrier thickness and the desired ‘target’ depth in oncological EMDA applications.

## 3. Pre-Clinical Studies and Clinical Applications of EMDA in Oncology

### 3.1. Urological Cancer

The first clinical application of EMDA in urology was in the delivery of local anesthetic agents for trans-urethral intra-vesical procedures [40]. The intact urothelium represents a highly impermeable barrier membrane that prevents the systemic absorption of ionized (Na^+^, K^+^, Cl^−^) and unionized (urea) solutes from the urinary system [7]. As such, EMDA in urology was devised to enhance locoregional drug delivery due to the limitations in passive transport.

Following a series of EMDA feasibility studies with local anesthetic drugs, its uro-oncologic application was first evaluated by Di Stasi et al. in in vitro studies, which compared passive versus electromotive delivery of MMC in the human bladder wall in the late 1990s [39,41]. In the preceding decade, intra-vesical MMC had been widely utilized amongst urologists following transurethral resection of non-muscle invasive bladder cancer (NMIBC) to prevent local recurrence, which can occur in up to 50% of patients [42]. By combining conventional locoregional MMC treatment with EMDA, Di Stasi’s group found that the application of 20 mA of pulsed direct current (DC) during 30 min resulted in increased penetration of MMC into all layers of the ex vivo bladder wall [39]. When EMDA (15 mA pulsed DC over 40 min) was applied to the bladders of adult mongrel dogs using an intravesical anode inserted through a Foley’s catheter, a similar EMDA-enhanced transport of dye materials into the submucosa and muscularis of the bladder wall was observed [43].

Subsequently, the applicability of EMDA with MMC was evaluated in a series of clinical studies (Table 1). Amongst them, three randomized controlled trials conducted by Di Stasi’s Italian group provided the highest quality of evidence supporting the use of EMDA in the adjuvant and neoadjuvant treatment of NMIBC [44,45,46]. In the first study (2003), 108 patients with early bladder cancer (Tis or T1) were randomized into three groups of 36 patients each following transurethral resection of bladder tumor (TURBT) and underwent (1) EMDA with MMC with 20 mA of electric current for 30 min or (2) passive MMC with a 1 h dwell time or (3) Bacillus Calmette-Guerin (BCG) with a 2 h dwell time, as part of adjuvant therapy [44]. Local and systemic adverse effects were monitored and included symptoms such as urinary frequency, fever, malaise and allergic reactions. The incidence of post-procedure bacterial or drug induced cystitis, prostatitis and epididymitis was also recorded. Comparing the three study arms, patients undergoing BCG treatment were more likely to experience local and systemic side effects. There was no difference in safety and toxicity profiles between passive and EMDA MMC groups. Treatment efficacy measured in terms of complete response and recurrence free rates were comparable in the BCG and EMDA MMC arms, but these were superior to passive MMC instillation. This led to the conduct of a second RCT in the adjuvant setting comparing the use of BCG alone with sequential BCG and EMDA with MMC [45]. Patients allocated to sequential BCG and EMDA with MMC experienced lower recurrence rates and had longer disease-free intervals.

In the neoadjuvant setting, EMDA with MMC was administered immediately prior to TURBT of NMIBC. The 3rd Italian RCT compared three patient groups: (1) TURBT alone, (2) immediate post-TURBT passive MMC and (3) pre-TURBT EMDA with MMC [46]. Irritative bladder symptoms were more common in groups 2 and 3 when compared with group 1 alone, and the use of EMDA immediately before TURBT did not result in higher rates of intra-operative complications. Recurrence and disease-free intervals were superior in the pre-TURBT EMDA group, although there was no difference in overall survival.

EMDA protocols adopted by the various centers show slight variations, with earlier studies using a lower current amplitude of 15 mA over shorter time periods of 20 min, while most subsequent studies utilize a maximum of 20 to 23 mA of pulsed DC applied over a duration of 30 min [47,48,49,50,51,52,53,54,55]. No differences in tolerability and adverse side effects were observed with the higher amplitude and longer durations. Amongst adjuvant studies, reported rates of complete response post-EMDA with MMC range from 40% to 82% while recurrence rates range from 14 to 52%. In addition, it was also found to have a potential role as salvage therapy in BCG refractory high grade NMIBC [55]. 

### 3.2. Skin Cancer

The skin is a highly heterogenous membrane and represents the largest organ in the human body. Its outermost layer, the stratum corneum (SC), is key to its barrier function that protects from the external environment [36]. The SC is approximately 15–20 µm thick and has a ‘brick-and-mortar’ structure comprising corneocytes supported by an inter-cellular matrix acting as an effective barrier that prevents the loss of water and the penetration of harmful compounds [56]. Beyond the SC, intercellular junctions, such as tight junctions and adherens junctions, also act as barriers to transdermal drug delivery [57].

In transdermal EMDA (often referred to as transdermal iontophoresis), a low amplitude of direct current, i.e., 0.5 mA/cm^2^, is used to enhance the penetration of a wide variety of drugs for the treatment of benign skin and sweat gland conditions, ulcers, scars and infections [36]. This should be differentiated from transdermal electroporation studies where high voltage pulses over short durations are applied to increase the permeability of the skin [22].

The use of EMDA in skin cancer has been described since the 1980s for the treatment of squamous cell carcinoma (SCC), basal cell carcinoma (BCC) and other skin tumors (Table 2). In these clinical studies, a variety of common chemotherapeutic agents such as cisplatin, 5-fluorouracil (FU), bleomycin and vinblastine were applied during EMDA. In general, a maximum of 4 mA (range 0.5 to 4 mA) of direct current is applied over the skin lesions during 10 to 30 min. Complete or partial response are common, especially after consecutive EMDA treatment of small, less aggressive lesions (e.g., BCC) [58,59,60,61,62,63].

In recent years, several preclinical studies have explored the use of novel anti-cancer agents and carriers in both melanoma and non-melanocytic skin cancers, i.e., BCC and SCC (Table 3). Doxorubicin, for example, is known to interact strongly with the SC, limiting its ability to be administered transdermally. However, when loaded in positively charged gel carriers during EMDA, improved penetration was achieved compared to water-based formulations. This may be explained by the interactions between the positively charged carrier or its degradation products with the negatively charged sites in the skin, such that the cationic doxorubicin is liberated from its SC binding sites and can penetrate into deeper layers of the skin [64]. Additionally, the use of nano-carriers such as liposomes, gold nanoparticles, dendrimers and lipid nanoparticles in conjunction with EMDA is increasingly explored to enhance the penetration of hydrophilic macromolecules [65,66,67,68,69,70,71,72,73]. When doxorubicin was encapsulated in solid lipid nanoparticles (SLNs) and EMDA applied, a 50-fold increase in penetration depth compared to passive diffusion of unformulated doxorubicin was achieved [65]. The doxorubicin-SLN treatment was also effective in inhibiting tumor cell survival and growth and was associated with increased keratinization and cell death in SCC murine models [65]. In a series of three studies by Labala and coworkers, gold nanoparticle carriers were used to optimize the transport of imatinib mesylate, anti-STAT3 SiRNAs and a combination of both using EMDA [68,71,72]. The co-delivery of both drugs was found to be feasible and resulted in greater suppression in STAT3 and a corresponding reduction in tumor cell viability in melanoma mice models as compared to single agent treatment alone. With increasing recognition of the efficacy of immunotherapy and anti-cancer vaccines in skin cancer, the application of EMDA with CpG-ODN (oligodeoxynucleotides containing unmethylated cytosine–phosphate–guanosine motifs) and cancer antigen gp-100 peptide KVPRNQDWL loaded in nanogels were also reported [66,67]. In both instances, improved penetration and a significant inhibition of tumor growth in melanoma mouse models were observed after EMDA. Similarly, in SCC xenograft models, transdermal EDMA application of EGFR-targeted immunoliposomes loaded with 5-FU was more effective than subcutaneous injection of the drug compound [73].

Photodynamic therapy (PDT) is another common treatment modality in skin cancers. It involves the topical application of a photosensitizer drug followed by illumination with visible light to activate the drug, which selectively destroys cancerous cells [74]. Common photosensitizing agents include 5-aminolevulinic acid (ALA, drug precursor), which penetrates the skin and is converted to protoporphyrin IX (active form) after exogenous application. In vitro studies evaluating ALA with EMDA have found that when 0.47 to 0.5 mA/cm^2^ of constant current is applied, the transport of ALA into the skin is significantly increased [75,76,77]. This effect was further enhanced when erbium: yttrium-aluminum-garnet (Erb:YAG) laser or microdermabrasion techniques were used to ablate the SC layer prior to EMDA [78]. However, as PDT was not applied in these experiments, the impact on cancer treatment is not known. However, when tetrasulfonated zinc phthalocyanine (ZnPcS4), a 2nd generation photosensitizer, was used in an EMDA experiment with PDT in SCC mouse models, a 15-fold improvement in tumor drug uptake and a 2.8-fold reduction in tumor volume were achieved [70].

### 3.3. Ophthalmic Cancer

Ocular EMDA has a long history and has been investigated since the early 1900s for the treatment of benign eye conditions such as corneal ulcers, keratitis and episcleritis and eye infections [17]. Trans-corneal or trans-scleral applications have been described. In the former, the EMDA device is placed over the cornea, and in the latter, the device is placed directly on the anterior sclera [79].

Currently, the evidence on the application of EMDA in ophthalmic cancer is limited to preclinical animal studies. The treatment of intra-ocular retinoblastoma with trans-scleral EMDA has been evaluated in mice and rabbits and a dose-dependent inhibition of tumor growth was observed with carboplatin [80,81]. During the experiments, electric currents ranging between 2.57 and 5.14 mA/cm^2^ were applied during 2 to 10 min. Optimal ocular drug transport was achieved with current densities between 2.57 and 3.85 mA/cm^2^, while a density of 5.14 mA/cm^2^ was associated with corneal–limbal toxicity [80]. However, when a hydroxyethyl methacrylate (HEMA) hydrogel carrier was used for carboplatin delivery during EMDA, no advantage over passive diffusion of the drug was seen [82].

### 3.4. Other Cancer Types

Trans-buccal EMDA in head and neck cancer, intra-ductal in breast tissue, and peri-pancreatic applications have been described [83,84,85,86,87]. In trans-buccal experiments, when bovine buccal mucosa was treated with 5-FU and leucovorin at 1 mA/cm^2^ during 20 min, EMDA resulted in a significant increase in mucosal penetration of both drugs compared to passive transport [80]. In a later study using nanoparticles as carriers for oxaliplatin, further improvements in penetration were observed, and this was accompanied by cytotoxic effects on SCC cells [84].

In canine and murine breast cancer models, a novel EMDA device was developed to administer an anti-estrogen drug (miproxifen phosphate, TAT-59) into mammary ducts via the nipple. The technique was feasible and resulted in significantly higher local drug concentrations when compared with oral administration [85]. In another series of studies evaluating the use of EMDA for the local delivery of cytotoxic therapies to both breast and pancreatic xenograft tumor models, Byrne et al. designed an implantable device that could be either implanted intra-abdominally or applied trans-dermally. Amongst orthotopic breast cancer mice, EMDA with cisplatin when directly applied to implanted tumors resulted in significant inhibition to tumor growth rates and extended life expectancy by 2-fold. A combination of intravenous with EMDA device cisplatin achieved the best outcomes [85]. Peri-pancreatic applications of gemcitabine and FOLFIRNOX (combination of 5FU, leucovorin, irinotecan and oxaliplatin) using the same EMDA device yielded significant tumor regression outperforming intravenous administration [87,88].

## 4. Future Outlook

### 4.1. Cellular Pathways Mediating EMDA

Traditionally, reported mechanisms for EMDA-mediated drug transport include electromigration, electroosmosis and electroporation (as described in Section 2). However, recent studies have found that cellular signaling pathways may play a role during EMDA. Susumu et al. evaluated the effect of faint electrical stimulus on intracellular events, including protein phosphorylation and changes in intracellular signaling factors and found that iontophoretic treatment induces Ca^2+^ influx into skin cells via a change in membrane potential. This in turn leads to a reduction in the expression of jap junction proteins and depolymerization of tight junction associated actin filaments, which enhances drug penetration [89].

In patients with glioblastoma, an emerging treatment is the use of tumor treating fields (TTFs), where low intensity, alternating currents are delivered to the tumor using electrodes placed in its vicinity [90]. TTFs have been shown to halt the cell cycle by disrupting spindle formation during metaphase leading to tumor cell death [91]. The use of TTFs in newly diagnosed glioblastoma was endorsed by the National Comprehensive Cancer Network (NCCN), with multiple ongoing clinical trials extending its use to other solid tumors [90,91,92,93]. Though conceptually different from EMDA-mediated drug transport, the success of TTFs in brain tumors highlights the potential for the use of electrical fields in various forms of anti-cancer treatment. Future studies should focus on evaluating the molecular and cellular mechanisms of EMDA and its anti-tumoral efficacy.

### 4.2. Extrapolating Current Evidence for Intra-Peritoneal Applications

The efficacy of intra-peritoneal (IP) drug delivery for the treatment of peritoneal cancers is limited by high intra-tumoral interstitial fluid pressure (IFP) that prevents effective drug penetration via diffusion and convection phenomena [5]. Current strategies aimed at improving IP drug delivery include the use of hyperthermia (i.e., hyperthermic intra-peritoneal chemotherapy, HIPEC), metronomic IP dosing, the use of novel drug carriers such as albumin, nanoparticles and hydrogels, and aerosolization of chemotherapeutic agents (i.e., pressurized intra-peritoneal aerosolized chemotherapy, PIPAC) [4,94,95,96]. The addition of electromotive forces during PIPAC in the form of electrostatic-PIPAC (e-PIPAC) has also gained increasingly popularity, with evidence suggesting improved spatial distribution and drug penetration (Figure 4). During e-PIPAC, up to 7.5 kV (current intensity < 10 µA) is applied to a stainless-steel brush electrode and results in a stream of negatively charged aerosolized drug particles that show accelerated delivery to the positively charged abdominal wall [94].

Given the success of EMDA in enhancing loco-regional drug delivery in the context of the urothelium, skin, sclera and tumoral tissues of varying origins, it may represent a promising solution to current challenges encountered during IP drug delivery. Drawing from our experience with e-PIPAC, we have established an in vitro EMDA IP model and evaluated the impact of applying a pulsed DC current to porcine peritoneal samples (Figure 5). Further in vitro and in vivo studies will be needed to confirm the safety, tolerability and efficacy in the context of peritoneal malignancies.

## 5. Conclusions

EMDA as a modality to improve loco-regional drug delivery is a rapidly growing field. Oncological applications are diverse and include the treatment of skin, eye and bladder cancer. Preclinical studies have also demonstrated its efficacy when combined with nanocarriers and novel drug delivery devices. Future efforts should focus on the clinical translation of EMDA in these fields as well as expanding its applicability to intraperitoneal drug delivery systems.

## Figures and Tables

**Figure 1 cancers-14-04980-f001:**
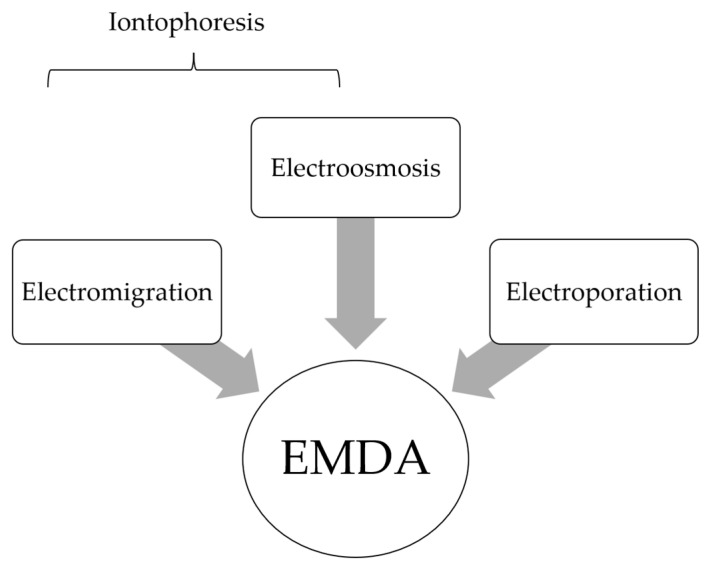
Electromotive drug administration (EMDA) encompasses the electrokinetic phenomena of electromigration (EM), electroosmosis (EO) and electroporation (EP). Conventionally, iontophoresis alone was commonly used to describe electric-driven drug transport.

**Figure 2 cancers-14-04980-f002:**
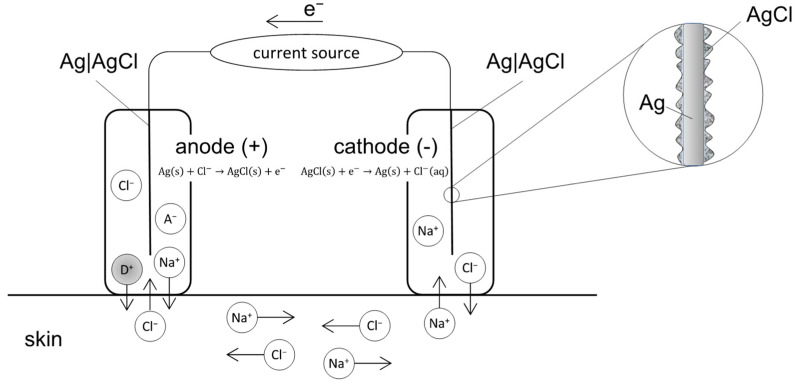
Schematic diagram of an iontophoretic device consisting of a current source and two Ag|AgCl electrodes. During EMDA, D+ is placed inside the electrode compartment bearing the same charge (i.e., the anode). Cations, including D+, are transported from the anode into the skin. At the same time, anions from the skin move into the anode. In the cathode, anions leave the cathode towards the skin, while cations move into the cathode.

**Figure 3 cancers-14-04980-f003:**
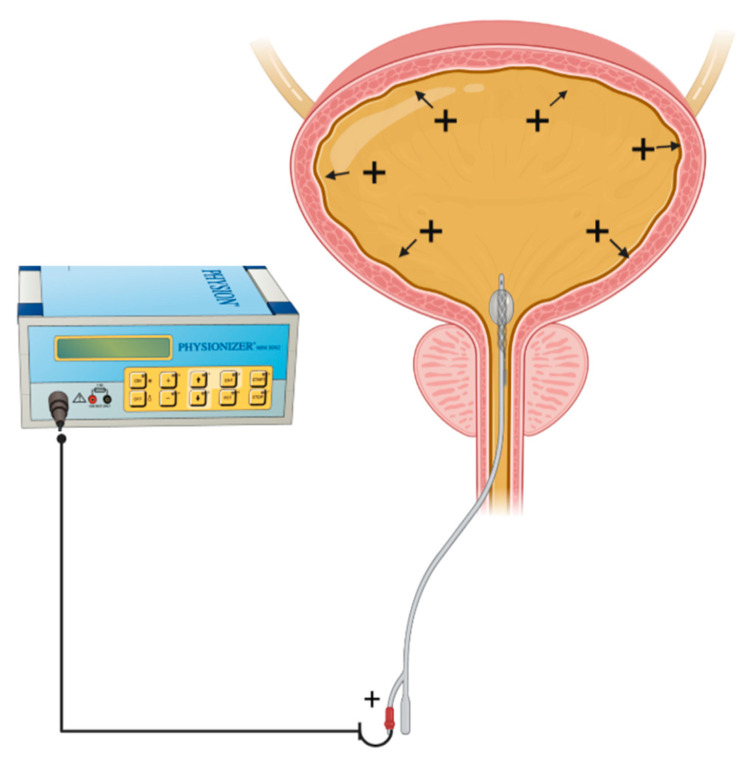
Schematic depiction of EMDA of a cationic drug molecule (+) within a filled bladder. The foley catheter contains a spiral Ag electrode at its tip and is connected to a current generator. During EMDA, grounding skin electrodes are placed over the anterior abdominal wall and connected to the cathode component of the generator.

**Figure 4 cancers-14-04980-f004:**
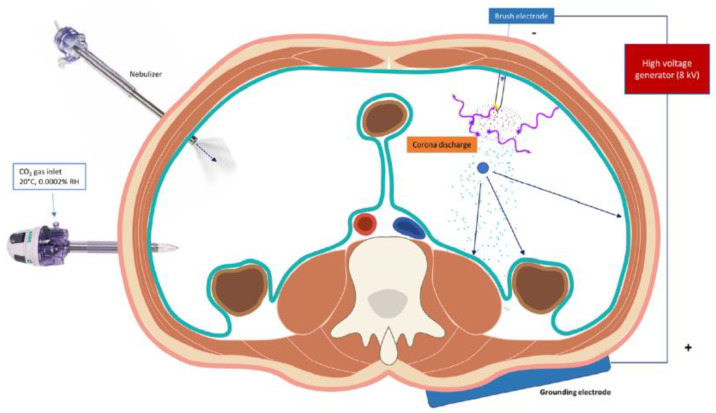
Schematic overview of electrostatic precipitation combined with pressurized intraperitoneal aerosol chemotherapy (e-PIPAC) where electromotive forces are used to improve IP drug distribution and penetration. (Adapted from Rahimi-Gorji et al., 2020).

**Figure 5 cancers-14-04980-f005:**
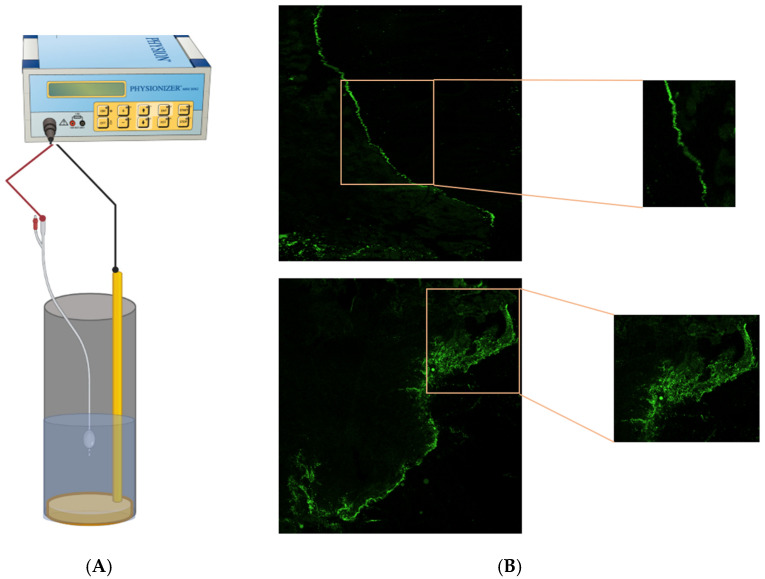
(**A**) Schema of in vitro IP model: custom-made plastic cylinder that mimics the abdominal cavity. A *Physionizer^®^ Mini 30N2* current generator is connected to a Foley catheter housing a silver spiral electrode and a grounding metallic plate. The setup aims to replicate the delivery of IP therapeutics. (**B**) Confocal microscopy images of porcine peritoneum tissue treated with fluorescent nanoparticles before (top) and after (bottom) EMDA, demonstrating improved penetration after EMDA. Experimental surgery lab, Ghent University, unpublished data.

**Table 1 cancers-14-04980-t001:** Clinical Studies on EMDA and treatment of non-muscle invasive bladder cancer (NMIBC).

Study/Y	Design	Clinical Context	Control Group		Intervention Group		Oncological Outcomes (Compare Control vs. Intervention)	
			No. of Patients	Treatment Regime	No. of Patients	Treatment Regime	Histological pCR	Recurrence and Survival
**Adjuvant Treatment in NMIBC**								
Brausi 1998 [47]	Multi-center Cohort	Primary or recurrent stage Ta or T1, Grade 1 or 2 TCC	12	40 mg PD MMC (2 h dwell time) weekly over 8 weeks	15	40 mg EMDA-MMC 15 mA over 20 min wkly over 8 wks	41.6% vs. 40%	RR: 60% vs. 33% DFI: 10.5 vs. 14.5 months
Riedl 1998 [48]	Single arm prospective	Ta or T1 TCC	NA	NA	22	40 mg EMDA-MMC 15 mA 20 min weekly over 4 weeks	56.6% (no control, at 14 months)	RR: 44%
Di Stasi 2003 [44]	Multi-center RCT	Ta or T1 TCC	36 36	A: 40 mg PD MMC (1 h dwell time) weekly over 6 weeks +/− additional course for non-responders B: BCG alone (2 h dwell time)	36	40 mg EMDA-MMC 20 mA 30 min weekly over 6 wks +/− additional course for non-responders	31% (Control A) vs. 58% (at 6 months)	RR: 75% and 52.8 (Controls A and B) vs. 52.8% DFI: 19.5 and 26 (Controls A and B) vs. 35 months *
Di Stasi 2006 [45]	Multi-center RCT	Primary or recurrent stage Ta or T1 TCC	105	BCG alone (2 h dwell time) weekly over 6 wks In CR, followed by BCG maintenance (to 10 months)	107	BCG (2 h dwell time) weekly over 2 weeks + 40 mg EMDA-MMC 20 mA 30 min weekly over 3 weeks In CR, followed by BCG + EMDA-MMC maintenance (to 10 months)	57.1% vs. 69%	RR: 57.9% vs. 41.9% * DFI: 21 vs. 69 months * DSS: 83.8% vs. 94.4% *
Gan 2016 [49]	Single arm prospective	High risk NMIBC	NA	NA	107	BCG (2 h dwell time) + 40 mg EMDA-MMC 20 mA 30 min weekly over 3 wks	71% (No control, at 1 year)	NR
Carando 2019# [50]	Retrospective single-arm	Intermediate and high risk NMIBC	NA	NA	65	40 mg EMDA-MMC 20 to 23 mA 30 min weekly over 8 weeks	82% (at 6 months)	NR
Carando 2020# [51]	Retrospective single-arm	Adjuvant setting Intermediate and high risk NMIBC	NA	NA	101	40 mg EMDA-MMC 20 to 23 mA 30 min weekly over 8 weeks Followed by EMDA-MMC maintenance (up to 12 months)	75% (at 6 months)	PFS (at 3 months): 94% PFS (at 6 months): 90%
Zazzara 2021# [52]	Retrospective cohort	Adjuvant setting Intermediate and high risk NMIBC	104	BCG alone (1.5 to 2 h dwell time) weekly over 6-weeks Followed by BCG maintenance (up to 12 months)	140	40 mg EMDA-MMC 20 to 23 mA 30 min weekly over 8 weeks Followed by EMDA-MMC maintenance (up to 12 months)	NR	RR: 20 vs. 14% (at maintenance phase)
**Neoadjuvant treatment in NMIBC**								
Colombo 2001 [53]	Single center Cohort	Ta or T1, Grade 1 or 2, <2 cm tumors	36	40 mg PD MMC (1 h dwell time) weekly over 4 wks	15	40 mg EMDA-MMC 15 mA 20 min weekly over 4 wks	27.7% vs. 40%	NR
Di Stasi 2011 [46]	Multi-center RCT	Neoadjuvant setting Ta or T1 TCC	116 119	A: TURBT alone B: TURBT + 40 mg PD MMC (1 h dwell time)	117	Pre-TURBT 40 mg EMDA-MMC 20 mA 30 min	NA	RR: 64% and 59% (Controls A and B) vs. 44% * DFI: 2 and 16 (Controls A and B) vs. 52 months *
Decae- Stecker 2018 [54]	Single-arm prospective	Neoadjuvant setting Primary or recurrent stage Ta or T1, tumor <2 cm	NA	NA	36	Pre-TURBT 60 mg EMDA-MMC 25 mA 25 min	25% (no control,1 year)	NR
**Other**								
Racioppi 2018 [55]	Single-arm prospective	Salvage bladder-sparing BCG-refractory High grade NMIBC	NA	NA	26	40 mg EMDA-MMC 20 mA 30 min weekly over 6 wks Followed by maintenance over 6 months	NA	High-grade disease-free rate 61.5%

PD passive diffusion; TCC Transitional Cell Carcinoma of the bladder; pCR pathological complete response; RR Recurrence rate; DFI Disease free interval; DSS Disease specific survival; NR not reported; NA Not applicable; RCT Randomized controlled trial; * significant *p* value; # Likely repeated cohort–same institution study with overlapping time periods.

**Table 2 cancers-14-04980-t002:** Clinical Studies on EMDA and treatment of skin cancer.

Study/Year	Design	Clinical Context	No. of Patients	Drug	EMDA Protocol	Oncological Outcomes
Luxenberg 1986 [58]	Case series	BCC or SCC (non-surgical candidates treated with systemic chemotherapy) Tumor site: Peri-orbital EMDA applied to recurrent or residual lesions after systemic chemotherapy	5	Cisplatin	0.5 to 1.5 mA DC over 20 min	PR in all patients
Chang 1993 [59]	Case series	BCC or SCC (non-surgical candidates) Tumor site: Face and others	12	Cisplatin	0.5 to 1.5 mA DC over 20 to 30 min Daily for 3–5 days or once/week	CR 26.7% of lesions, PR 46.7%, minimal response 26.7% Factors associated with CR: (1) Lesion <3 cm (2) BCC (3) Daily administration
Bacro 2000 [60]	Case report	BCC Tumor site: Lower extremity	1	Cisplatin	Daily for 5 days over 4 weeks	CR was achieved
Welch 1997 [61]	Case series	SCC-in-situ Tumor site: Head and Neck, Trunk, Upper and lower extremities	26	5-FU	4 mA DC over 10 min Twice/week over 4 weeks	CR in 96% (25 out of 26) patients
Tsuji 1991 [62]	Case study	Verrucous carcinoma (not surgical candidate) Tumor site: Lips	1	Bleomycin	2 mA DC over 30 min Trice/week over 2 weeks	CR was achieved Recurrence-free at 6 months
Smith 1992 [63]	Case series	Kaposi’s sarcoma in HIV patients Tumor site: Upper and lower extremities, face, tongue	4	Vinblastine sulfate	4 mA DC over 10 min once	CR in 29% of lesions, PR in 71%

DC Direct Current BCC basal cell carcinoma SCC squamous cell carcinoma PR partial response CR complete response FU Fluorouracil HIV Human immunodeficiency virus.

**Table 3 cancers-14-04980-t003:** Pre-clinical studies on EMDA and treatment of skin cancer.

Study/Year	Study Type	Experimental Tissue/Skin Cancer Type	Drug Used	Carrier (if Any) or Carrier Solution	EMDA Protocol	Outcomes
Taveira 2009 [64]	In vitro	Porcine skin Melanoma (Murine cells)	Doxorubicin (cationic)	Chitosan gel (cationic), water, HEC gel (non-ionic)	0.5 mA/cm^2^ over 6 hrs	Doxorubicin with chitosan gel carrier result in improved epidermal penetration (in porcine skin) with EMDA when compared with passive diffusion (PD) 3-fold increase in doxorubicin cytotoxicity with EMDA in cancer cells over PD
Huber 2015 [65]	In vitro and In vivo	Porcine skin SCC (Murine model)	Doxorubicin (cationic)	Solid lipid nanoparticles (SLN), water	0.5 mA/cm^2^ over 6 h	2-fold improvement of doxorubicin penetration (in porcine skin) with SLN carrier Significantly improved SCC tumour inhibition using doxorubicin-SLN with EMDA over PD
Kigasawa 2011 [66]	In vivo	Melanoma (Murine model)	CpG-ODN	NaCl	0.3 mA/cm^2^ over 1 h	Improved penetration and distribution of drug in epidermis and dermis with EMDA over PD Significant inhibition of tumour growth with local or distant application of drug with EMDA compared with non-treated controls
Toyoda 2015 [67]	In vitro and In vivo	Porcine Skin Melanoma (Murine model)	Cancer-antigen gp-100 peptide	PEG modified Nanogels	0.4 mA/cm^2^ over 1 h	Significant increase in no. of Langerhans cells in epidermis with gp100 and EMDA over non-treated controls Significant tumour growth suppression with antigen peptide-loaded nanogels and EMDA over non-treated controls
Labala 2015 [68]	In vitro	Porcine skin Melanoma (Murine cells)	Imatinib mesylate	Gold nanoparticles	0.47 mA/cm^2^ over 4 h	6.2-fold improvement of drug-nanocomplex penetration with EMDA over PD Significant inhibition of tumour cell viability by nanocomplex over non-treated controls
Venuganti 2015 [69]	In vitro and In vivo	Porcine skin Skin tumor (Murine model)	Anti-sense Oligonucleotide (ASO)	PAMAM Dendrimer	0.3 mA/cm^2^ over 4 h	Increased penetration of ASO-Dendrimer complex into epidermis with EMDA over PD Significant apoptosis of skin tumour with drug complex over non-treated controls
Lemos 2016 [70]	In vivo	SCC (Murine model)	ZnPcS_4_ * (anionic)	HEC Gel (Non-ionic)	0.5 mA/cm^2^ over 30 min	15-fold improvement of drug uptake by tumour with EMDA over PD 2.8-fold reduction in tumour volume in the group of mice treated with photosensitizer with EMDA followed by photodynamic therapy compared with non-treated controls
Labala 2016 [71]	In vitro	Porcine skin Melanoma (Murine cells)	anti-STAT3 SiRNA	Gold nanoparticles (Chitosan capped)	0.47 mA/cm^2^ over 4 h	Complex was able to successfully penetrate skin with EMDA Nanoparticle complex inhibited cell growth by 50%
Labala 2017 [72]	In vitro and In vivo	Melanoma (Murine cells) Melanoma (Murine model)	Co-delivery of anti-STAT3 SiRNA and Imatinib mesylate	Gold nanoparticles	0.5 mA/cm^2^ over 2 h	Combination of 2 drugs resulted in greater suppression of STAT3, reduction in tumour cell viability when compared with single drug delivery Transdermal EMDA delivery of nanocomplex resulted in significant reduction in tumour volume, weight, and STAT3 suppression (similar to direct intra-tumoral injection) over non-treated controls
Petrilli 2018 [73]	In vitro and In vivo	Porcine skin SCC (Murine model)	5-FU	EGFR-targeted immunoliposome	0.5 mA/cm^2^ over 6 h	2-fold penetration of drug in skin with immunoliposomes with EMDA compared with control liposomes 2-fold reduction in tumour volume with transdermal 5-FU containing liposomes and EMDA compared with subcutaneous injection

SCC squamous cell carcinoma; PD passive diffusion; HEG Hydroxyethyl cellulose; ALA aminolaevulinic acid; HEPES (4-(2-hydroxyethyl)-1-piperazineethanesulfonic acid)–zwitterionic sulfonic acid buffering agent; Erb:YAG erbium:yttrium-aluminium-garnet; PEG Polyethylene glycol; CpG-ODN unmethylated; CpG motifs 5-FU Fluorouracil; EGFR epidermal growth receptor; SiRNA small interfering RNA PAMAM poly (amidoamine). ZnPcS4 tetrasulfonated zinc phthalocyanine * photosensitizer.

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
