# Peer review of "Electromotive Enhanced Drug Administration in Oncology: Principles, Evidence, Current and Emerging Applications"

_cancers, 2022, doi:10.3390/cancers14204980_

Round 1

Reviewer 1 Report

This is a well-written review paper containing principles, influencing factors and applications of EMDA. It is a topic of interest to the researchers in the related areas, but for the benefit of the reader, certain statements require further justification and some small points need to clarify. My detailed comments are as follows:

(1) As a significant concern in clinical applications, methods to avoid thermal damage to healthy tissues need to be listed or discussed in part 2.2, like special designs for EMDA devices. You described Rahimi-Gorji’s work in Figure 4, the generator provided a high voltage at 8 kV, but the current was only 10 μA, the principles need to be discussed.

(2) More examples are needed to show relationships between drug physicochemical properties and EMDA except for Gangarosa’s work in part 2.3.

(3) The format of tables in this paper is needed to be optimized, some descriptions need to be consistent. Before submitting a revision be sure that your materials are properly prepared and formatted.

Author Response

This is a well-written review paper containing principles, influencing factors and applications of EMDA. It is a topic of interest to the researchers in the related areas, but for the benefit of the reader, certain statements require further justification and some small points need to clarify. My detailed comments are as follows:

(1) As a significant concern in clinical applications, methods to avoid thermal damage to healthy tissues need to be listed or discussed in part 2.2, like special designs for EMDA devices. You described Rahimi-Gorji’s work in Figure 4, the generator provided a high voltage at 8 kV, but the current was only 10 μA, the principles need to be discussed.

Thermal damage is prevented by capping the maximum current density to 0.5mA/cm2  , through the use of well-saturated absorbent pads and ensuring that there is no direct contact between metallic components and the skin during transdermal EMDA. In intra-vesical EMDA, applied current is capped at 20mA to prevent thermal damage and this has been proven safe in multiple phase 2 clinical trials as elaborated in the section 3.1 of the manuscript.

In Figure 4, we described the application of electrical forces in electrostatic pressurized intra-peritoneal aerosolized chemotherapy, e-PIPAC. While extrapolating the concept of electromotive forces to enhance  IP-delivery is a promising strategy, the ideal current amplitude, duration, and other parameters will have to be further optimized in in-vitro and in-vivo studies before considerations for clinical applications. The authors have elaborated on this aspect in section 4.

(2) More examples are needed to show relationships between drug physicochemical properties and EMDA except for Gangarosa’s work in part 2.3.

The authors have augmented these principles with more examples in section 2.3.

(3) The format of tables in this paper is needed to be optimized, some descriptions need to be consistent. Before submitting a revision be sure that your materials are properly prepared and formatted.

All tables have been reformatted for ease of reading in the submitted revision.  

Reviewer 2 Report

In this review article Wong Si Min et al, outline the principles and factors that 16 govern electrically driven drug transport. I find this review of high overall merit and clinical significance and I would be happy to support its publication, if the authors address a couple of questions/concerns.

1. The introduction section and historical context is well-written and documented, gradually introducing the reader to this specialized topic.

2. The theoretical analysis is adequate but over specialized at places. I am concerned that if the reader is not a biophysics expert, will have a hard time to follow the context. The authors are highly suggested to simplify this section of the manuscript.

3. I high appreciate the fact that the authors have collected all the active clinical trials to date. This goes a long way to support the clinical importance of the method. Nevertheless, I would the author to discuss whether the EMDA method would change the Tumor Microenvironment leading to altered response to immunotherapy, a targeted therapy that has been approved for bladder cancer at this point. Information on any of this topic would be of a great clinical value.

4. Last but not least, the authors do not discuss the molecular pathways that EMDA alters or leverages in order to achieve better drug delivery. The authors should discuss whether the EMDA creates neo-epitopes or altered immune infiltrating cellular populations in tumor micro-environment, leading to anti tumor effect by its self.

I would like to wish best of luck to the authors during their revision process.

Author Response

Reviewer 2

In this review article Wong Si Min et al, outline the principles and factors that  govern electrically driven drug transport. I find this review of high overall merit and clinical significance and I would be happy to support its publication, if the authors address a couple of questions/concerns.

1. The introduction section and historical context is well-written and documented, gradually introducing the reader to this specialized topic.

2. The theoretical analysis is adequate but over specialized at places. I am concerned that if the reader is not a biophysics expert, will have a hard time to follow the context. The authors are highly suggested to simplify this section of the manuscript.

We agree that the section on fundamental  principles of EMDA may be overly theoretical & have simplified the section & moved the relevant materials into supplementary text. We have also shortened section 2.1 on EMDA devices.

3. I high appreciate the fact that the authors have collected all the active clinical trials to date. This goes a long way to support the clinical importance of the method. Nevertheless, I would the author to discuss whether the EMDA method would change the Tumor Microenvironment leading to altered response to immunotherapy, a targeted therapy that has been approved for bladder cancer at this point. Information on any of this topic would be of a great clinical value.

4. Last but not least, the authors do not discuss the molecular pathways that EMDA alters or leverages in order to achieve better drug delivery. The authors should discuss whether the EMDA creates neo-epitopes or altered immune infiltrating cellular populations in tumor micro-environment, leading to anti tumor effect by its self.

I would like to wish best of luck to the authors during their revision process.

There are only a handful of publications describing potential cellular pathways mediating EMDA in in-vitro settings. This has been added to section 4. Unfortunately, it is not known if EMDA results in alteration of the tumor microenvironment & this will definitely be an interesting area to look into moving forward.

Reviewer 3 Report

This review discusses the use of electrocution in cancer. It provides background and theory, plus the current clinical and pre-clinical evidence. Strengths of the review include a good flow and compelling presentation of the subject matter. The review could be strengthened as follows:

Major Points

1. Along with the electric current driving drug delivery, electric current has impacts on neuronal responses in tissues, including secretion of disease-modifying compounds. Potential contributions of these other bioelectric responses to clinical successes should be discussed in the context of drug delivery.

2. Limitations, failures, and challenges of EMDA need better discussion.

3. Explanation of the state of regulatory approval/clinical guidelines for these interventions in major jurisdictions (eg EU, US, UK) would be a helpful measure of the success of these approaches.

Minor points

1. Table 1 is challenging to read. Cleaning up the headings and providing more width may help.

2. line 202 ‘non-ionized’ is used, but ‘unionized’ is used everywhere else.

3. Line 217-218 “corneocytes embedded in intracellular lipids” does not make sense. Do the authors mean extracellular lipids?

Author Response

 This review discusses the use of electrocution in cancer. It provides background and theory, plus the current clinical and pre-clinical evidence. Strengths of the review include a good flow and compelling presentation of the subject matter. The review could be strengthened as follows:

Major Points

1.Along with the electric current driving drug delivery, electric current has impacts on neuronal responses in tissues, including secretion of disease-modifying compounds. Potential contributions of these other bioelectric responses to clinical successes should be discussed in the context of drug delivery.

2. Limitations, failures, and challenges of EMDA need better discussion.

High voltage electrical stimulations have been described in its applications to targeted organs e.g. skeletal muscle, brain tumours (through tumor treating fields, TTF). These have been describe to result in changes in neuronal response and secretion of disease modifying compounds.  However, these phenomena have not been described in the context of EMDA and definitely warrants further evaluation in future studies.

In EMDA, drug transport is mediated through iontophoresis and electroporation (defined by the formation of aqueous pores) where the movement of charged drug ions in an electrolyte solution is enhanced through the application of a low amplitude current. This differs from high voltage & direct application of electrical stimulations.

3. Explanation of the state of regulatory approval/clinical guidelines for these interventions in major jurisdictions (eg EU, US, UK) would be a helpful measure of the success of these approaches.

This information has been augmented in section 2.2 on EMDA related devices.

Minor points

1. Table 1 is challenging to read. Cleaning up the headings and providing more width may help.

2. line 202 ‘non-ionized’ is used, but ‘unionized’ is used everywhere else.

 3. Line 217-218 “corneocytes embedded in intracellular lipids” does not make sense. Do the authors mean extracellular lipids?

 We have implemented format changes in table 1 & have standardized the terms “unionized” versus “non-ionized” used in the manuscript.

With regards to #3, the stratum corneum has been reported to have a “bricks-and-mortar” structure where the corneocytes represent the bricks and the intercellular lamellar lipid membrane represents the mortar [ref 33]. We have amended to provide better clarification.